# Seasonal Phenology of the Major Insect Pests of Quinoa (*Chenopodium quinoa* Willd.) and Their Natural Enemies in a Traditional Zone and Two New Production Zones of Peru

**Luis Cruces** [1,2,*], **Eduardo de la Peña** [3]  **and Patrick De Clercq** [2]

1   Department of Entomology, Faculty of Agronomy, Universidad Nacional Agraria La Molina, Lima 12-056, Peru
2   Department of Plants & Crops, Faculty of Bioscience Engineering, Ghent University, B-9000 Ghent, Belgium; patrick.declercq@ugent.be
3   Department of Biology, Faculty of Science, Ghent University, B-9000 Ghent, Belgium; Eduardo.DeLaPena@ugent.be
*   Correspondence: luiscruces@lamolina.edu.pe; Tel.: +051-999-448427

**Abstract:** Over the last decade, the sown area of quinoa (*Chenopodium quinoa* Willd.) has been increasingly expanding in Peru, and new production fields have emerged, stretching from the Andes to coastal areas. The fields at low altitudes have the potential to produce higher yields than those in the highlands. This study investigated the occurrence of insect pests and the natural enemies of quinoa in a traditional production zone, San Lorenzo (in the Andes), and in two new zones at lower altitudes, La Molina (on the coast) and Majes (in the "Maritime Yunga" ecoregion), by plant sampling and pitfall trapping. Our data indicated that the pest pressure in quinoa was higher at lower elevations than in the highlands. The major insect pest infesting quinoa at high densities in San Lorenzo was *Eurysacca melanocampta*; in La Molina, the major pests were *E. melanocampta*, *Macrosiphum euphorbiae* and *Liriomyza huidobrensis*; and in Majes, *Frankliniella occidentalis* was the most abundant pest. The natural enemy complex played an important role in controlling *M. euphorbiae* and *L. huidobrensis* by preventing pest resurgence. The findings of this study may assist quinoa producers (from the Andes and from regions at lower altitudes) in establishing better farming practices in the framework of integrated pest management.

**Keywords:** quinoa; *Eurysacca melanocampta*; *Macrosiphum euphorbiae*; *Liriomyza huidobrensis*; *Frankliniella occidentalis*; natural enemies; IPM; Peru

## 1. Introduction

In the Andes of Peru, quinoa has mostly been cultivated as a staple crop by smallholders, with limited resources that do not allow them to use advanced agricultural technology. In this ecoregion, small-scale farming has largely been practiced, characterized by low inputs, the restricted use of machinery and rain-fed irrigation [1,2]. However, in the last years, as a consequence of the increasing demand for quinoa on the international markets and the resulting export boom and crop expansion, farmer associations have been created. In turn, this has led to improvements in crop management by the incorporation of agricultural machinery and technical assistance [3]. The production of this Andean grain in the highlands is mostly organic, with a relatively low yield level that is partially compensated by the higher market price as compared with conventional quinoa [4,5].

This revalorization of quinoa motivated many farmers in the Andes to shift from staple crops (such as potato, corn and legumes) to quinoa but also gained attention of growers from regions at lower

altitudes (i.e., from the "Maritime Yunga" to the coastal areas) [2,5,6]. In these newly exploited areas, small-, medium- and large-scale cultivation is practiced, characterized by the implementation of relatively advanced farming techniques including technified irrigation (especially in areas belonging to local irrigation projects such as "Majes-Siguas" and "Olmos" in the Arequipa and Lambayeque departments, respectively) and the use of machinery, pesticides, fertilizers and, in some cases, modern equipment for spraying [4,7,8]. Therefore, the production of quinoa in these areas is mainly conventional, with higher yield levels than in the highlands [1,4,5].

A relatively long list of phytophagous insects has been reported to infest quinoa in the Andean areas [7,9]. However, only the quinoa moths *Eurysacca melanocampta* (Meyrick) and *Eurysacca quinoae* Povolný (Lepidoptera: Gelechiidae) are considered of major importance, while other herbivorous species, including thrips and aphids, are generally considered of minor relevance [10,11]. For the non-traditional areas of quinoa production, pest communities infesting the crop also include *E. melanocampta*, as well as polyphagous insects such as the aphid *Macrosiphum euphorbiae* (Thomas) (Hemiptera: Aphididae), the thrips *Frankliniella occidentalis* Pergande (Thysanoptera: Thripidae), the leafminer fly *Liriomyza huidobrensis* (Blanchard) (Diptera: Agromyzidae) and the hemipteran pests *Nysius simulans* Stål (Hemiptera: Lygaeidae) and *Liorhyssus hyalinus* (Fabricius) (Hemiptera: Rhopalidae) [9]. Knowledge about the economic impact of the latter pests on quinoa production in the newly exploited areas is, however, still scarce.

In this context, the present study aimed to explore the seasonal occurrence of the relevant insect pests on quinoa in two new production zones as compared to a traditional production area, by analysing their incidence in the crop, as a function of the presence of their natural enemies, environmental factors and the farming practices specific to each region. The findings of this study should be of interest for local quinoa growers for improving their pest management practices and also for other farmers who intend to explore new areas for quinoa production in Peru and other countries that share similar pest complexes.

## 2. Materials and Methods

### 2.1. Field Sites

The study was carried out in three areas of Peru: a traditional quinoa production zone (San Lorenzo; 11°50′33″ S, 75°22′45″ W, 3322 m above sea level [m a.s.l.]) located in the Andean region, and two non-traditional quinoa production areas, one located on the coast (La Molina; 12°06′ S, 76°57′ W, 244 m a.s.l.) and the other in the "Maritime Yunga" region (Majes; 16°21′31″ S, 72°17′16″ W, 1410 m a.s.l.) (Figure S1).

The monitored fields were cultivated under conventional farming practices. The field sites in the localities of La Molina and San Lorenzo belong to the experimental and production fields of the National Agrarian University La Molina, whereas the field site assessed in Majes belongs to a private farmer. The characteristics of each field site and the cultivation and pest management specifications are given in Table 1. Meteorological data for the three localities can be found in Supplementary Figure S2.

**Table 1.** Growing specifications for quinoa during the sampling period in the localities of La Molina, Majes and San Lorenzo (Peru).

| | Localities | | |
|---|---|---|---|
| | **La Molina District, Province of Lima, Department of Lima** | **Majes District, Province of Caylloma, Department of Arequipa** | **San Lorenzo District, Province of Jauja, Department of Junín** |
| Mean monthly temp. (minimum–maximum) | 16.67–22.97 °C | 10.52–25.52 °C | 6.96–20.06 °C |
| Mean monthly RH (minimum–maximum) | 74.65%–96.25% | 31.2%–60.2% | 65.51%–75.75% |
| Total precipitation during the sampling period | 5.9 mm | 0 mm | 276.2 mm |
| Sowing–harvest | 2 September 2015–10 January 2016 | 15 May 2016–20 September 2016 | 11 January 2016–20 May 2016 |
| Field dimensions | 85 m × 96.3 m (0.66 ha) | 93.5 m × 96.3 m (0.9 ha) | 102 m × 96 m (0.98 ha) |
| Variety | Pasancalla | Inia Salcedo | Pasancalla |
| Irrigation | Surface irrigation 100 irrigation furrows of 85 cm width, 10 irrigation blocks | Drip irrigation 110 irrigation furrows of 85 cm width, 4 irrigation blocks | Rain-fed 120 furrows of 85 cm width, 12 irrigation blocks |
| Soil type | Clay loam | Loamy sand | Loam |
| Neighbouring crops | Quinoa (*Chenopodium quinoa*); barley (*Hordeum vulgare*); kiwicha (*Amaranthus caudatus*); wheat (*Triticum* spp.) | Quinoa (*Chenopodium quinoa*); artichoke (*Cynara scolymus*) | Quinoa (*Chenopodium quinoa*); corn (*Zea mays*); potato (*Solanum tuberosum*) |
| Fungicides | 1° benomyl (15 September 2015); 2° metalaxyl + mancozeb (4 October 2015); 3° dimethomorph (20 October 2015); 4° propamocarb + fluopicolide (3 November 2015) | 1° benomyl (22 May 2016); 2° metalaxyl + mancozeb (12 June 2016); 3° dimethomorph (26 June 2016); 4° propamocarb + fluopicolide (10 July 2016) | 1° benomyl (2 January 2016); 2° metalaxyl + mancozeb (14 February 2016); 3° dimethomorph (28 February 2016); 4° propamocarb + fluopicolide (15 March 2016) |
| Insecticides | 1° *Bacillus thuringiensis* (27 October 2015); 2° dimethoate + methomyl (3 November 2015); 3° emamectin benzoate + methomyl (8 December 2015) | 1° alpha-cypermethrin (22 May 2016); 2° emamectin benzoate (29 May 2016); 3° zeta-cypermethrin (12 June 2016); 4° alpha-cypermethrin (26 June 2016); 5° alpha-cypermethrin + emamectin benzoate (10 July 2016) | 1° *Bacillus thuringiensis* + emamectin benzoate (4 April 2016) |
| Weed management | Manual control | Manual control | Manual control |
| Previous crop | Wheat | Corn | Fallow period of 6 months |

Source for meteorological data: The weather station "Von Humbold" at the National Agrarian University La Molina, the weather station Map-Pampa de Majes of the National Service of Meteorology and Hydrology of Peru (SENAMHI), and the weather station at the Regional Institute of Highland Development in Jauja of the National Agrarian University La Molina.

### 2.2. Sampling Procedure

The sampling campaign was performed considering the planting season for each location, and samples were taken evenly throughout the crop phenology, from two weeks after germination to one week before harvest. In La Molina, 15 samplings were performed from 22 September 2015 to 29 December 2015; in Majes, 10 samplings were performed from 26 May 2016 to 12 September 2016; and in San Lorenzo, 9 samplings were performed from 31 January 2016 to 12 May 2016. The lower number of samplings executed in Majes and San Lorenzo as compared to La Molina was due to the lesser accessibility of the first two sites.

At each location, the field was divided into 5 sectors (considering the slope of the field and the irrigation blocks); in each sector, 5 quinoa plants, at least 20 m apart, were sampled (Figure 1). Each sampled plant was cut at its base and placed into a container with water, alcohol and some drops

of liquid detergent. After taking five plants per sector, they were carefully chopped into small pieces, and the whole sample (including the liquid content) was transferred to a labelled, airtight container to be transported to the laboratory for further processing. Plants from borders were always avoided for sampling. During collection, care was taken to minimize the disturbance of any insects present on the plant.

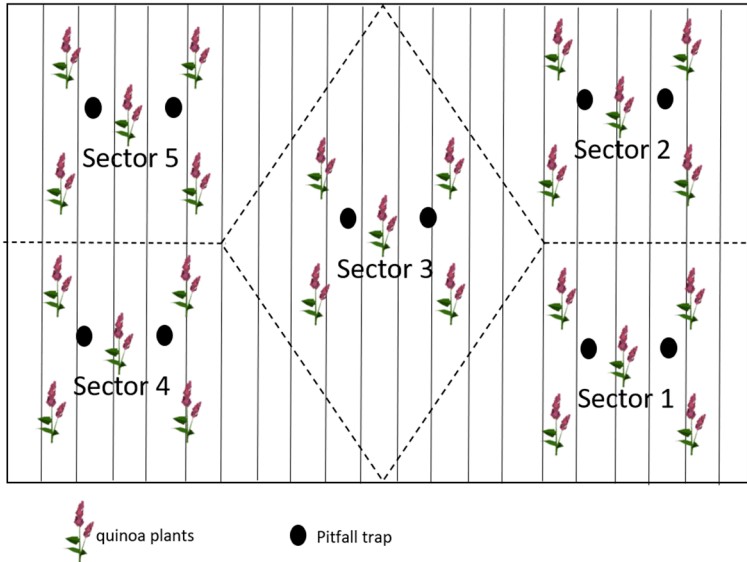

**Figure 1.** Sectorization and sampling scheme applied to the monitored fields. Transversal lines represent the direction of the furrows.

To complement the analysis, the epigeous insects were examined throughout the crop phenology with ten pitfall traps (transparent, ⌀ 10 cm, 10% ethylene glycol, water and detergent) and 2 traps per sector (Figure 1), which were left during the whole crop phenology (from one week after germination to one week before harvest). The pitfall trap content was periodically collected on the same day when the quinoa plants were sampled.

### 2.3. Sample Processing and Identification

All samples were processed at the laboratories of the Museum of Entomology "Klaus Raven Büller" of the National Agrarian University La Molina, in Lima, Peru, where the collected specimens were deposited.

The recipients containing the sampled plants and pitfall trap samples were poured onto a 1 mm mesh sieve and carefully washed with water, removing larger materials, except for the leaves with mines; these were later examined under a binocular stereoscope (Carl Zeiss, Stemi 508 LAB, Zeiss, Jena, Germany) to remove the leafminer larvae and/or their parasitoids. The remaining samples (i.e., the collected insect specimens) were transferred to labelled glass vials containing 75% *v/v* ethanol for conservation and further processing (i.e., identification).

The specimens were sorted on the basis of morphological characteristics as morphospecies. For the hemimetabolous insects, adults and nymphs were taken into account, but for holometabolous insects, only the harmful stages (larvae and/or adults) were considered in the study. For the aphids, mummified specimens were also considered, to calculate the parasitism level based on the number of parasitized aphids and the total number of aphids collected. For the leafminers, the parasitism level was calculated based on the number of parasitoids and leafminer larvae extracted from the mines.

When feasible, the most relevant morphospecies (taking into account abundance and functional behaviour) were identified at the genus and species levels, with the help of taxonomic keys and morphological descriptions provided in the literature as follows: for Aphididae spp. [12,13],

Aphidiinae spp. [14–17], *Allograpta exotica* (Wiedemann) [18], *Blennidus peruvianus* (Dejean) [19–22], *Diabrotica sicuanica* Bechyne [23], *Epitrix* spp. [24], Eulophidae genera [25], *E. melanocampta* [26], *Geocoris* spp. [27], *Halticoptera* sp. [28], *Heterotrioza chenopodii* (Reuter) [29], *L. hyalinus* [30,31], *L. huidobrensis* [32,33], *Nabis capsiformis* Germar [34], *N. simulans* [35] and *Russelliana solanicola* Tuthill [36,37].

Molecular tools were applied for identifying and/or confirming the species *Lysiphlebus testaceipes* (Cresson), *Aphidius matricariae* Haliday, *Aphidius colemani* Viereck, *Aphidius rosae* Haliday, *Aphidius avenae* Haliday, *Aphidius ervi* Haliday, *F. occidentalis*, *L. huidobrensis*, *L. hyalinus*, *M. euphorbiae* and *Rhopalosiphum rufoabdominale* (Sasaki). DNA extraction and PCR procedures were performed in the Laboratory of Agrozoology, Department of Plants and Crops at Ghent University, Belgium, following specific protocols provided in the literature [38–42]. Specimens of *Epitrix* sp., *Macrosiphum* sp., *Myzus* sp., *Therioaphis* sp., *Geocoris* sp., *Chrysocharis* sp., *Halticoptera* sp., *Diglyphus* sp. and *Closterocerus* sp. could not reliably be identified at the species level, either morphologically (since this is only confirmed by a specialist of the corresponding taxa) or based on molecular methods.

Expert taxonomists assisted by identifying and/or confirming certain species: *H. chenopodii* and *R. solanicola* were identified by Daniel Burckhardt from the the Naturhistorisches Museum of Switzerland; the dolichopodids were identified by Daniel Bickel from the Australian Museum; *Astylus subannulatus* Pic was identified by Robert Constantin from the Entomological Society of France; *N. simulans* was identified by Pablo Dellapé from the Museo de La Plata in Argentina.

### 2.4. Data Analysis

For the most relevant species (major pests and their natural enemies), curves of seasonal occurrence were built to analyse the pest–natural enemy interactions, which were interpreted in the context of each scenario (i.e., the environmental factors and the agricultural practices at each field site).

The statistical analyses were performed using the R software, version 3.4.2 [43] (packages: vegan, agricolae, and MASS) [44–46].

For the population comparisons, a one-way ANOVA was applied to the data after having tested the normality and homoscedasticity through Shapiro–Wilk and Bartlett tests, respectively. When the data did not meet the assumption of the homogeneity of variances, the Box–Cox transformation method was used to stabilize the variances. When the ANOVA was significant, Tukey's honestly significant difference test was used to compare the groups. All the tests were analysed at the significance level of $\alpha = 0.05$.

## 3. Results

### 3.1. Abundance and Diversity of Phytophagous Insects

The plant samplings throughout the crop phenology at the field site in La Molina yielded 24 morphospecies of phytophagous species, among which *M. euphorbiae*, *E. melanocampta*, *F. occidentalis*, *L. huidobrensis* and *H. chenopodii* encompassed 99.1% of the total abundance of herbivorous insects. At the field site in Majes, 12 morphospecies of phytophagous insects were found, including *F. occidentalis*, *Myzus* sp. and *Macrosiphum* sp., encompassing 99.2% of the total abundance of herbivorous insects. The hemipteran pests *L. hyalinus* and *N. simulans*, which were recently reported to be causing severe damage in newly exploited areas for quinoa production [7,8], were found at low densities at these two localities. Finally, in San Lorenzo, 16 morphospecies of phytophagous insects were found, with *F. occidentalis*, *E. melanocampta*, *Myzus* sp., *Macrosiphum* sp. and *H. chenopodii* accounting for up to 97.3% of the total abundance of herbivores. At this locality, *A. subannulatus*, *D. sicuanica* and *Epitrix* sp., which are mentioned in the literature as minor pests of quinoa [7,10,23], were collected in very small numbers.

Rank–abundance curves of phytophagous insects were built as a function of their abundance in the samplings at each field site (Figure 2). Comparatively, the curve for the San Lorenzo field site (SL) has a

less pronounced slope than the curves for the other sites. This suggests that the phytophagous species are more evenly distributed at this locality or there was a lower dominance of the most abundant pests as compared to at the La Molina and Majes field sites, which were characterized by a higher dominance of certain taxa.

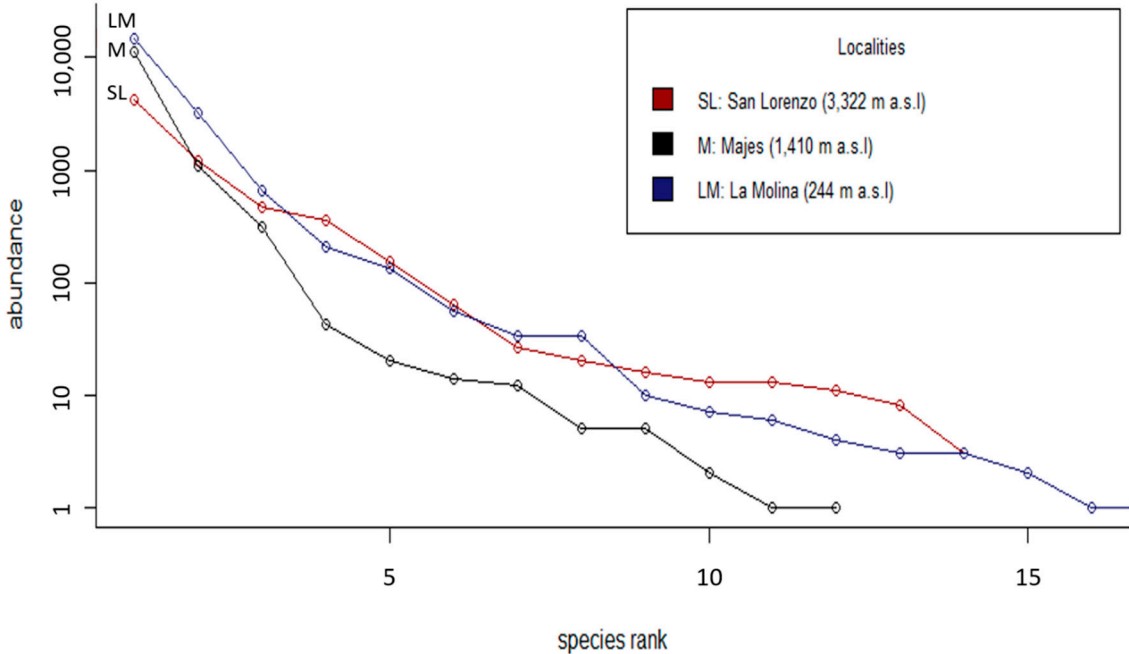

**Figure 2.** Rank–abundance curve of the phytophagous insects that infested the quinoa crop in San Lorenzo, Majes and La Molina (log series distribution).

### 3.2. Phenology of Phytophagous Insects of Economic Importance

### 3.2.1. Quinoa Moth

At the field site in La Molina, the seasonal occurrence curve of *E. melanocampta* (Figure 3A), based on the number of larvae per plant, had two peaks throughout the crop phenology. The first peak occurred on 3 November 2015, with an average of 7.9 individuals per plant; this was controlled with the insecticide treatment dimethoate + methomyl (Table 1), from which the pest later resurged. The second peak occurred on 8 December 2015, with up to 65.6 specimens per plant on average; this infestation was managed with emamectin benzoate + methomyl, leading to a marked suppression of this pest. The first spraying with *Bacillus thuringiensis* var. *kurstaki* performed on 27 October 2015 against a low population of this moth had little effect.

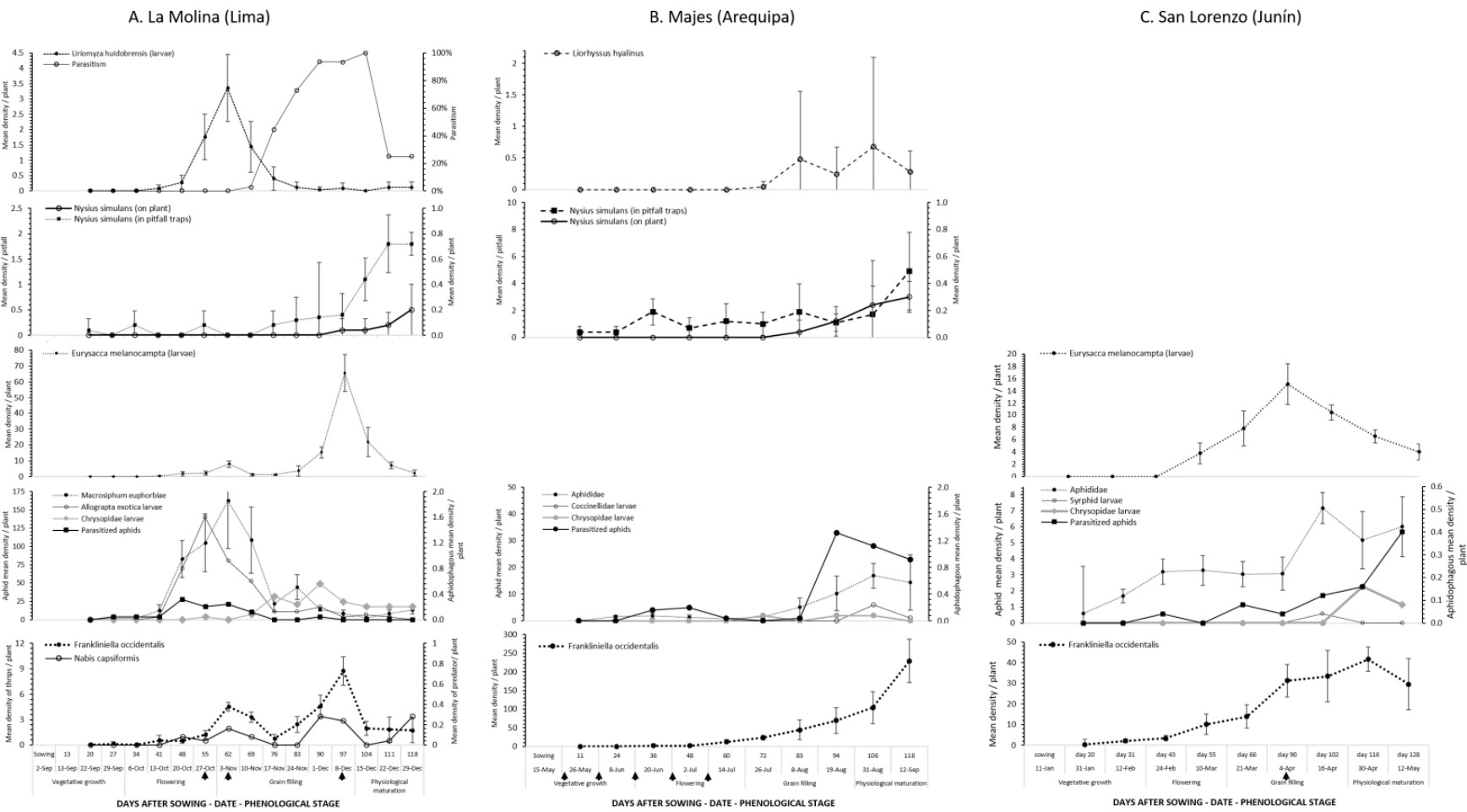

**Figure 3.** Seasonal incidence of the main insect pests (mean number per plant or pitfall trap ± SD) and their associated natural enemies (mean number per plant or percent parasitism) sampled on quinoa at the field sites in (**A**) La Molina, Lima (from 22 September 2015 to 29 December 2015); (**B**) Majes, Arequipa (from 15 May 2016 to 12 September 2016); and (**C**) San Lorenzo, Junín (from 31 January 2016 to 12 May 2016). Arrows on the time axis indicate the timing of the insecticide applications.

Caterpillars of this species were scarcely observed in Majes, likely due to the constant treatments with broad-spectrum insecticides during the first 60 days of the cropping season.

At the field site in the traditional quinoa production locality, San Lorenzo, the occurrence of *E. melanocampta* larvae had its maximum number on 4 April 2016 (Figure 3C). The caterpillars started to infest the plants 43 days after sowing (24 February 2016) and progressively increased in number up to 15.1 larvae per plant, on average. At this point, they were controlled with emamectin benzoate + *B. thuringiensis* var. *kurstaki*, which efficiently reduced the larval incidence thereafter.

With regard to the environmental variables (Figure S2), in San Lorenzo, the rain had a notorious effect on the establishment of this moth in the field, since the infestation began only after the raining period had finished (at the end of February). Minimum temperatures that mostly ranged between 0 and 10 °C likely also had an effect on the moth, slowing down its incidence. Contrarily, precipitation at the locality of La Molina was scarce, and the temperature was quite stable throughout the cropping season, with small differences between the maximum and minimum; thus, the interaction between these environmental factors and *E. melanocampta* incidence was not evident. Additionally, no specialized natural enemies of *E. melanocampta,* such as parasitoids, were observed during the sampling campaign, either at La Molina or at San Lorenzo.

The mean density of *E. melanocampta* larvae sampled on the plants at La Molina and San Lorenzo over the total sampling period was compared. After applying the Box–Cox transformation method ($\gamma = -0.5$) to the data, the ANOVA indicated that the overall larval density was significantly higher in La Molina than in San Lorenzo ($F_{1,8} = 31.46$, $p < 0.001$).

### 3.2.2. Aphid–Natural Enemy Complex

The infestation by aphids at the field sites was related to more than one species: At the locality of La Molina, a high incidence of *M. euphorbiae* (99.2%) and scarcely any *R. rufiabdominale* (0.2%) were found; in Majes, *Myzus* sp. (77.3%) and *Macrosiphum* sp. (22.7%) were observed; and in San Lorenzo, the aphid complex consisted of *Myzus* sp. (55.7%), *Macrosiphum* sp. (42.8%) and *Therioaphis* sp. (1.5%).

The seasonal occurrence curve of *M. euphorbiae*, based on the number of aphids per plant, had two peaks in La Molina (Figure 3A). The first occurred on 3 November 2015, with the highest recorded population (162.3 individuals per plant on average), promoting the development of sooty mould on the leaves as a consequence of their honeydew secretion; this infestation was controlled efficiently with methomyl + dimethoate. The second peak occurred on 24 November 2015 (with 44.8 specimens per plant on average), but at this point, no insecticide was used, so the corresponding reduction of the aphid population in the following days may, in part, be explained by the action of the natural enemies, especially chrysopid larvae, the population of which increased in this period.

According to seasonal changes in the aphid abundance in La Molina, a temporal succession in the numerical response of the aphidophagous guilds was observed (Figure 3A). Larvae of the predatory syrphid *A. exotica* first appeared, with peak numbers in the early developmental period of the aphid population, followed by aphidiine wasps but with a maximum parasitism level of only 2.5%; at the later phases of the crop, chrysopid larvae were found again. Wasps of the Aphidiinae complex collected in the pitfall traps consisted of *L. testaceipes* (Cresson), *A. matricariae* and *A. colemani*.

In Majes, the incidence of Aphididae was very low during the first 60 days after sowing (15 September 2016–14 July 2016), probably due to the intensive insecticide treatments applied in the early stages of the crop. From then onwards, the infestation continuously grew, reaching up to 22.5 individuals per plant on average (on 31 August 2016), followed by a decrease that may, in part, be explained by the action of predators such as chrysopid and coccinellid larvae, and parasitism by Aphidiinae wasps (Figure 3B). When examining the specimens belonging to this group collected in the pitfall traps at Majes, the complex was formed by *A. colemani*, *A. ervi*, *A. avenae* and *A. rosae*.

Contrarily to the field site in La Molina, syrphids were absent in Majes, and the most abundant aphidophagous group was the Aphidiinae wasp complex. These appeared in the early stages of the crop, but their establishment became more significant after the period of insecticide treatments,

during the grain formation and maturation, with a maximum parasitism level of 13.5%. Coccinellid and chrysopid larvae appeared in small numbers, also at the end of the crop phenology (Figure 3B).

At the field site in San Lorenzo, the incidence of the Aphididae was considerably lower than in La Molina, amounting to only 7.1 specimens per plant, on average (Figure 3C). Given this low infestation, no pesticide treatment was applied against the aphids and the spraying with emamectin benzoate + *B. thuringiensis* targeted against *E. melanocampta* larvae had no visible effects on the Aphididae. Based on the number of aphid specimens sampled per plant, there was a quite stable population density until 84 days after sowing (4 April 2016), followed by a slight increase.

When juxtaposing the environmental variables (Figure S2) and the aphid occurrence, only in San Lorenzo can a certain interaction be observed: for example, the aphid establishment at the beginning of the crop phenology only prospered when the rains subsided; also, the large differences between the maximum and minimum temperatures and chilling conditions in the period from 28 April 2016 to 4 May 2016 coincided with a decrease in the aphid population. These factors may also have affected the abundance of the natural enemies since only a single larva of Syrphidae and six larvae of Chrysopidae were collected throughout the crop phenology, and the maximum parasitism level reached no more than 7.2% during the cropping season (Figure 3C). In this locality, *A. colemani* and *Aphidius* sp. were recorded in the pitfall traps.

The mean overall densities of Aphididae at the three localities were compared. After applying the Box–Cox transformation method ($\gamma = 0.1$) to the data, the ANOVA indicated that there were highly significant differences between the localities ($F_{2,12} = 146.4$, $p < 0.001$). Tukey's HSD test indicated that the aphid density in La Molina was significantly higher than in San Lorenzo ($p < 0.001$) and Majes ($p < 0.001$), the latter locality having a significantly higher aphid incidence than San Lorenzo ($p = 0.033$).

### 3.2.3. Leafminer Flies and Natural Enemy Complex

Adults and larvae of *L. huidobrensis* were found in considerable abundance only in La Molina, and therefore, the seasonal occurrence of this species was analysed in detail only for this locality. Since the adults of leafminer flies are very active and easily disturbed, they could not be efficiently sampled by way of the plant sampling, and therefore, the collected adult data were excluded from analysis.

The seasonal occurrence of *L. huidobrensis* had a maximum number of 3.3 larvae per plant (Figure 3A). This infestation level was reduced by the treatment with methomyl + dimethoate targeted against aphids on 3 November 2015. Later, the parasitoid complex, formed mainly by eulophids and pteromalids [47], had an important role in decreasing the leafminer population, with parasitism reaching up to 100% (Figure 3A).

When examining the specimens collected in the pitfall trap sampling, the following leafminer fly parasitoids were recorded: two species of Pteromalidae (*Halticoptera* sp.1 and *Halticoptera* sp.2) and seven of Eulophidae (*Chrysocharis* sp.1, *Chrysocharis* sp.2, *Diglyphus* sp.1, *Closterocerus* sp.1, *Cirrospilus* sp.1 and two non-identified taxa). From this complex, *Halticoptera* sp.1 and *Chrysocharis* sp.2 were present in markedly larger numbers than the others.

### 3.2.4. Hemipteran Pests

The rhopalid *L. hyalinus* was only collected in the non-traditional quinoa production localities La Molina and Majes, but in small numbers. In the first locality, only six specimens of this species were found, in the last plant sampling. In Majes, the population size was greater and focused in the grain filling stage (Figure 3B), although the mean density of this bug on the plants never surpassed 0.68 specimens per plant, with a large standard deviation, suggesting that the spatial distribution of this species in the crop is not uniform but clumped.

The lygaeid *N. simulans* was also collected only at the localities of La Molina and Majes. Since this species has a primarily soil-surface-dwelling behaviour, the seasonal occurrence was analysed, contrasting the population found on the plants with the specimens collected in the pitfall traps.

In La Molina, the population of *N. simulans* at ground level was characterized by a considerable increase from the grain filling stage onwards, and the insect started to inhabit the plants around the physiological maturation stage (Figure 3A). The field eventually had a strong outbreak of this bug from the harvest cut to the day of threshing; unfortunately, the population size at that time could not be recorded because the last sampling was performed one week before cutting. Since the cut plants were lying on the soil surface during 10 days for drying, this greatly favoured the infestation of quinoa by *N. simulans*.

In Majes, the occurrence of *N. simulans* at the soil level remained low until the grain filling stage, when the bugs also started to infest the plant; from then onwards, the population constantly increased, reaching up to 4.9 individuals per pitfall trap, on average, in the last sampling. On the plant, the population size remained small, reaching only 0.32 individuals per plant, on average, in the last sampling (Figure 3B).

### 3.2.5. Western Flower Thrips

The seasonal occurrence curve of *F. occidentalis* in La Molina was characterized by two peaks (Figure 3A). The first occurred on 3 November 2015, reaching only 4.5 individuals per plant on average, but the infestation was likely reduced by the insecticide treatment (methomyl + dimethoate) targeted against the aphids and *E. melanocampta*. The second peak occurred on 8 December 2015, reaching 5.2 individuals per plant on average, whereafter the thrips incidence was likely reduced by the insecticide treatment (methomyl + emamectin benzoate) applied to control *E. melanocampta*. These pesticide sprayings may have obscured the interactions between the thrips and certain generalist natural enemies such as *N. capsiformis* and chrysopids found in the samplings.

The seasonal occurrence curve of *F. occidentalis* in Majes had an exponential shape, reaching up to 198 thrips per plant on average, in the last sampling. The population at the early stage of the crop phenology was small, probably due to the intensive use of insecticide during this phase. Thereafter, the infestation had a continuous increase, suggesting that there were few restrictive factors for the population growth during the monitored period; thus, natural enemies such as chrysopid larvae appeared to have had little effect on the thrip infestation (Figure 3B).

The seasonal occurrence of *F. occidentalis* in San Lorenzo had a maximum number of up to 41.7 thrips per plant on average (Figure 3C). It is likely that the minimum temperatures between 28 April 2016 and 4 May 2016, with values going down to 0 °C, had a detrimental effect on this pest (Figure S2).

The mean densities of the *F. occidentalis* per plant sampling at the three field sites were compared. After applying the Box–Cox transformation method ($\gamma = 0.1$) to the data, the ANOVA indicated that there were highly significant differences between the localities ($F_{2,12} = 226.8$, $p < 0.001$). Tukey's HSD test showed that the thrips density in Majes was overall significantly higher than in La Molina ($p < 0.001$) and San Lorenzo ($p < 0.001$); the density at the latter site was significantly greater than at La Molina ($p < 0.001$).

## 4. Discussion

The survey at the field in San Lorenzo confirmed the relevance of *E. melanocampta* for quinoa in the Andes of Peru, which is deemed, in the literature, to be the crop's key pest [10,48,49]. Likewise, the findings in La Molina shed light on the importance of this moth at the coastal level, a newly exploited region for quinoa production [7], and revealed that polyphagous insects such as *M. euphorbiae* and *L. huidobrensis* may infest quinoa plants in high densities. Nonetheless, similar observations could not be made in Majes, where pest insects were scarcely collected in the early stages of the crop, likely due to the pest management scheme (Table 1), and only the population of the cosmopolitan pest *F. occidentalis* prospered in high densities when the insecticide sprayings stopped.

In the highlands of Peru, most of the cultivated quinoa is rain-fed irrigated. For this reason, farmers only cultivate the crop during the raining season, being forced to have a fallow period [1].

In this context, *E. melanocampta* may have two generations in the Andean region [50]; the first occurs between November and December in early sowings, and the second is between March and April for late sowings, the latter coinciding with the period during which this moth infested the crop in San Lorenzo. In Majes and La Molina (like other coastal areas), farmers do not depend on the rain for irrigation, and they can sow quinoa at almost any time, so several generations of this moth may develop throughout the year in these valleys. Under this pattern of *E. melanocampta* incidence, designing pest management strategies for quinoa in the Andes is more feasible than in the non-traditional quinoa production zones, such as Majes and La Molina, unless farmers of the latter valleys take into account the organization of their sowing periods when setting up integrated pest management (IPM) schemes.

To better understand the impact of the incidence of *E. melanocampta* at the studied field sites, we refer to the economic threshold level of 3 to 15 larvae per plant, as suggested in previous studies [51,52]. Whereas in San Lorenzo, the infestation by this pest reached levels of up to 15 larvae per plant in 40 days (from 24 January 2016 to 4 April 2016), in La Molina, by only 21 days (from 17 November 2015 to 8 December 2016), even higher levels were attained (with up to 65 larvae per plant on average), exceeding, by far, the said threshold. According to Villanueva [52], the occurrence of 30 larvae per quinoa plant may cause a 58.8% yield loss, whereas 70 larvae per plant could lead to an 85% loss.

One environmental factor that likely played a key role for *E. melanocampta* infestation is temperature. Previous observations pointed out that the pest's biological cycle is shortened from 75 to 28 days as the temperature increases from 20 to 24 °C [50]. In San Lorenzo, the mean monthly temperature oscillated between 14.4 and 15.3 °C, with large differences between the maximum and the minimum (up to 18 °C on average), which may have slowed down the development of the moth. Conversely, in La Molina, where the mean monthly temperature ranged from 19.4 to 21.6 °C (with maxima of up to 29.4 °C), the differences between the maximum and minimum temperatures did not exceed 7 °C, meeting the conditions for this pest to develop more generations throughout the cropping season; this may explain, in part, the higher incidence at this location as compared to San Lorenzo.

Aphids are considered secondary or occasional pests of quinoa in the Andes of Peru and Bolivia [49], probably because their damage has been hard to pin down in terms of yield reduction or economic losses due to their overall low population density in the fields [53]. The environmental variables in the highlands are often unfavourable for their population build up (i.e., rains, chilling temperatures and large differences between the minimum and maximum temperatures). For example, in San Lorenzo, the minimum temperature during the cropping season dropped to 0.1 °C, which is detrimental to aphid populations, which are considered in the chill-susceptible group, with "pre-freeze mortality" being the dominant cause of death at low temperatures [54]. Contrariwise, the field site in La Molina had favourable conditions of temperature and relative humidity for the aphids to thrive (with up to 162 specimens per plant on average) [55]. With respect to Majes, the intensive use of insecticides during the first stages of the crop phenology and low incidence of the aphids at later stages did not allow revealing any such relation between climate and aphid populations.

Quinoa harbours an important diversity of natural enemies [9], including aphidophagous insects [11]. However, this beneficial fauna is likely also affected by the unfavourable climate in San Lorenzo or the intensive insecticide treatments in Majes. These conditions appeared to have impaired the predatory group to a somewhat higher degree than the parasitoids, given that Aphidiinae wasps were collected in these two localities with parasitism levels of up to 13.5% in the first locality and 6.1% in the second, whereas the aphidophagous predators in San Lorenzo were scarce, and in Majes, they only developed once the pesticide spraying had finished. These observations could be explained, in part, due to the fact that the developed larvae of parasitoids inside the host integument are, to some degree, protected from pesticide sprays, and part of the population inside the aphid mummy stage may experience a functional refuge [56].

In La Molina, more aphidophagous insects (in terms of abundance) were found than in the other two localities. A temporal succession in their occurrence was observed, which is related to their degree of feeding specialization: the aphid specialists (Aphidiinae wasps and predatory syrphid

larvae) appeared in the early stages of infestation by *M. euphorbiae*, whereas the more generalist Chrysopidae larvae appeared at later stages [57–59]. The effectiveness of these natural enemies, however, was likely perturbed by the insecticide applications. For example, the first spraying at 55 days after sowing with *B. thuringiensis* to control *E. melanocampta* may have had detrimental effects on *A. exotica* larvae, given that after this treatment, the increasing trajectory of their seasonal occurrence curve shifted to a decreasing trend, with a population reduction of around 42%. Although Horn [60] found, on collards, that aphidophagous Syrphidae were reduced by a treatment with *B. thuringiensis* var. *kurstaki*, more studies are needed to clarify the potential risks of the use of *B. thuringiensis* for syrphid larvae.

The second treatment at the field site in La Molina with the insecticides dimethoate and methomyl was also detrimental to the syrphid larval population, likely due to both direct toxicity [61] and a reduction in its aphid prey populations. Larval populations of chrysopids appeared after this insecticide treatment; being the predominant aphid predators at the later stages of the crop, they may have played an important role in keeping the aphids at a low density for some time after this spraying.

Thrips are also considered to be a secondary pest of quinoa, and there are no substantiated reports of significant yield reductions [53,62]. However, the seasonal occurrence patterns of *F. occidentalis* observed in Majes suggested that under favourable conditions, the thrips may infest the crop in an exponential way, reaching high levels of up to 191 thrips per plant on average. Considering that *F. occidentalis* possesses the basic characteristics for the fast development of pesticide resistance (a short generation time, high fecundity and haplodiploid breeding system) [63], and pyrethroid insecticides are being widely used in Majes [8], it is warranted to monitor the development of resistance in local populations of *F. occidentalis* to insecticides belonging to this chemical group. This would allow the implementation of proper insecticide resistance management by local farmers.

*L. huidobrensis* is another polyphagous pest that infested quinoa at relatively high densities (up to 3.36 larvae per plant) at the La Molina field site at mid stage of the crop phenology. The insecticide treatment on 9 November 2015 with dimethoate + methomyl markedly reduced the leafminer infestation. In the later stages of the crop, the temperature may have become less favourable (reaching up to 29 °C), preventing the pest from resurging. Previous studies indicate that high temperatures (25–30 °C) negatively influence the oviposition capacity of *L. huidobrensis* [64]. Conversely, the parasitoid complex of *L. huidobrensis* appears to be favoured by this range of temperatures [65–67]. Consequently, the seasonal occurrence of the parasitoids might have led to an effective control of the leafminer populations, with up to 100% parasitism (as the season became warmer), preventing *L. huidobrensis* from resurging. The occurrence of the parasitoid species in the field followed a similar pattern as in previous observations in potatoes in La Molina, where *Halticoptera* and *Chrysocharis* were the most abundant genera and, sporadically, *Diglyphus*, *Closterocerus* and *Ganaspidium* species were collected [65].

*L. hyalinus* and *N. simulans* have been reported as infesting quinoa in large numbers in the departments of Lambayeque and Lima at the coastal level and in Arequipa in the "Maritime Yunga" region of Peru [7,8]. These hemipteran pests were observed causing severe damage to quinoa in the last months of 2013, throughout 2014 and in the first semester of 2015, during which some farmers admitted the overuse of pesticides even during the grain maturation stage [8]. Although no high level of infestation was registered in the present study, vigilance should be maintained, particularly when considering that the nymphs and adults of these true bugs cause direct damage to the grains by their piercing–sucking feeding habit during the grain filling and maturation stages, when management by applying insecticides increases the risk of residues on the harvested grains.

Producers may not be aware of *N. simulans* during the first stages of the crop because of its terrestrial behaviour, cryptic appearance and minute size. Moreover, the traditional way of harvesting quinoa, which involves leaving the cut plants on the ground for drying before threshing, favours *N. simulans* infestation. Another factor that promotes the pest's incidence is its numerous host plants, encompassing a variety of crops and weeds, that allow them to find food in a wide variety of habitats [7].

The strategy of pest control applied by the farmer at the field site in Majes followed a fixed schedule of treatments rather than a system based on the infestation level (the two first sprayings being performed every 7 days after sowing and the remaining three treatments, every 14 days). These insecticide applications occurred only during the first 60 days of the crop phenology, in order to reduce the risks of harvests being contaminated with chemical residues (E. Falconi, personal communication, May 2016, Majes). This management scheme appears to be used by most of the local quinoa growers, including also the recurrent use of pyrethroids [8]. This practice may be positive in terms of obtaining grains without residues, but the continuous use of active ingredients with the same mode of action (i.e., alpha-cypermethrin and zeta-cypermethrin) may eventually lead to the development of pesticide resistance in some of the key pests [68,69]. Besides, the excessive use of broad-spectrum pesticides such as pyrethroids could cause harm to the environment [70] and have a negative impact on the natural enemy complex in quinoa [71].

Conversely, the insecticide use in San Lorenzo was more appropriate, given that the treatments were performed once the pest reached a certain threshold. Besides, selective insecticides (*B. thuringiensis* + emamectin benzoate) were applied in a single treatment to control *E. melanocampta*. Nonetheless, this scheme does not reflect the general use of chemicals by farmers in the highlands growing conventional quinoa, who mainly use pesticides of the synthetic pyrethroid and organophosphate types [4,8,49]. Likewise, at the field site in La Molina, the pesticide treatments were also based on the infestation level of the pests; here, however, a mix of selective and non-selective insecticides were applied at a very high level of infestation. The pest management strategies deployed in the three localities suggest the continued need for agricultural extension programmes in order to improve the use of agrochemicals.

## 5. Conclusions

The present study examined the occurrence of the major insect pests of quinoa and their natural enemies in a traditional production zone in the Andean region (San Lorenzo), and two non-traditional areas for quinoa production in Peru at lower elevations (La Molina, on the Coast, and Majes, in the "Maritime Yunga" ecoregion). The data gathered by on-plant and pitfall sampling show that the pest pressure in quinoa is higher at the lower altitudes than in the highlands of Peru. Although there are better conditions in the non-traditional quinoa production zones for attaining higher yields than in the Andean region, pests are likely to become an important barrier for successful quinoa production, a situation that may worsen if pesticides are incorrectly used. These are issues that farmers from Peru, and other South American countries, will eventually face when exploiting new production areas. Studies on the biology and ecology of the key species of pests and their natural enemies will aid in implementing suitable pest control strategies for the crop. Particularly, additional studies are needed to clarify the potential risks of aphids and *F. occidentalis* for quinoa production, especially in the non-traditional zones.

**Supplementary Materials:** The following are available online at http://www.mdpi.com/2077-0472/10/12/644/s1. Figure S1: Localities of La Molina, San Lorenzo and Majes in the map of Peru, Figure S2: Fluctuation of the daily mean temperature (maximum and minimum) and daily precipitation during the sampling period in La Molina–Lima, San Lorenzo–Junín and Majes–Arequipa.

**Author Contributions:** Conceptualization, L.C., E.d.l.P. and P.D.C.; methodology, L.C. and P.D.C.; investigation, L.C., E.d.l.P. and P.D.C.; formal analysis, L.C., E.d.l.P. and P.D.C.; data curation, L.C.; writing—original draft preparation, L.C.; writing—review and editing, L.C., E.d.l.P. and P.D.C.; supervision, E.d.l.P. and P.D.C. All authors have read and agreed to the published version of the manuscript.

**Funding:** This research was funded by THE PROJECT 2: "Development of Value Chains for Biodiversity Conservation and Improvement of Rural Livelihoods"—Sub Project: Native Grains, VLIR-UOS IUC/UNALM.

**Acknowledgments:** We thank Daniel Burckardt from the Naturhistorisches Museum of Switzerland for confirming the identity of the psylloids and Pablo Dellapé from the Museum of La Plata in Argentina for confirming the identity of *Nysius simulans*. We also thank the professors from the National Agrarian University La Molina in Peru; Luz Gómez, chief of the Cereals and Native Grain programme; and Clorinda Vergara, chief of the Museum

of Entomology "Klaus Raven Büller" for the facilities and permits. Finally, we acknowledge VLIR-UOS/UNALM for funding this study.

**Conflicts of Interest:** The authors declare no conflict of interest.

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
