# Peer review of "Seasonal Phenology of the Major Insect Pests of Quinoa (Chenopodium quinoa Willd.) and Their Natural Enemies in a Traditional Zone and Two New Production Zones of Peru"

_agriculture, doi:10.3390/agriculture10120644_

Round 1

Reviewer 1 Report

The title, abstract and introduction chapters adequately introduce the studies conducted. The methodology (including statistical analysis) are presented with rigor and are adequate for the studies performed. The authors could make a more in depth statistical analysis of the diversity of insects among regions, eventually with the computing of a diversity Index, or a similar tool to more rigorously compare the entomological diversity between the three studied zones.

The Results chapter is complete, factual, and presents adequate Tables and Figures, although in Figures 2-4 I suggest to add information on the timing of the insecticide applications (with small arrows on the exact date, or something similar), allowing the reader to better understand the rope of the insecticide applications on the demographic patterns of insect pests and their natural enemies. In Table 1 I do not think that presenting the “Mean monthly precipitation” is useful for the reader, ultimately the “Total precipitation during the sampling period” information would be much more interesting and informative.

The discussion is well structured and informative; the authors correctly interpret their data, although some precaution is recommended when analyzing the “apparent” peaks of the main pests, as some of the peaks are artificial outcomes of the insecticide treatments, which reduced the insect´s populations in a time when they were in fact growing to a (future) peak which was never reached… Such is the case of the first peak (I would not consider this a peak for the reasons exposed before) of E. melanocampta in Figure 2 (and other examples), and this should be more clearly referred in the Discussion chapter.

Although a little out of the scope of the manuscript, I´m intrigued on the long-term sustainability of growing a crop which requires irrigation on (semi)-desertic regions, maybe this could be briefly be discussed on the Introduction or Discussion Chapters…

The references are adequate, updated, and complete.

A few additional comments/questions:

  • At least for the most important pests, I suggest presenting the scientific classifications (Order, Family) the first time they´re mentioned, for the readers not familiarized with the most important Quinoa pests;
  • Please place in Italics the scientific names of the Genus/species on line 122 onwards;
  • Line 157, the title should be: “Abundance and diversity of phytophagous insects”
  • In Line 262, the date presented differs from the date in Table 1 for the same treatment;
  • Species already mentioned in the text don´t need to be presented in full scientific name once more (lines 331, 335, etc);
  • In Figure S1, the locations are not clear with black-and-white printings, please replace the color circles by different shapes for each region;

Reviewer 2 Report

This paper results interesting, mainly because the authors have worked on a plant more and more cultivated. My only suggestion is the following: the authors should present the graphics in a comparable way. I tried to put together the figures in such a way the results are more comparable, and I observed some differences in the phenology of insects (e.g., the peak of aphids does not match with the phenological stage of Quinoa in all the sites. It is probable that by showing the results in this way, the authors will be able to demonstrate better differences in the interaction plant/insects.

Please, check at the end of the paper, where the new figure is shown.

Small comments are highlighted in the text.
